# Piezomagnetoelectric effects in a candidate Kitaev magnet

**Vilmos Kocsis** [1] ✉, **Sven Luther**[2], **Nicolás Pérez**[1], **Weiliang Yao**[3], **Hannes Kühne** [2], **Anja U. B. Wolter** [1], **Yuan Li** [3,4] & **Bernd Büchner** [1,5,6]

The exactly solvable Kitaev model with its frustrated bond-dependent interactions has attracted enormous attention due to its exotic physics hosting fractional spin excitations as well as its promising prospects for quantum information technology. However, there is no pristine realization of the Kitaev model due to the significant Heisenberg and off-diagonal exchange interactions. While these additional exchange interactions are considered as obstacles on the route towards the desired Kitaev quantum spin liquids, the interplay between these magnetic anisotropies and the Kitaev interaction has lead to numerous intriguing phenomena. Here we demonstrate a new phenomenon, the coexistence of the Kitaev interaction with the piezomagnetoelectric effect (simultaneous magnetoelastic and magnetoelectric responses), which can offer electric field driven manipulation of the ground state and the fractional spin excitations. Our study reports the direct observation of the magnetoelectric (ME) effect in a Kitaev-Heisenberg, the quantum spin liquid candidate $Na_2Co_2TeO_6$, and highlights the magnetoelastic response as a sensitive gauge of phase transitions. We discuss that the ME effect originates from the *pd*-hybridization mechanism, which allows local polarization independently from any magnetic order. This mechanism can transfer the frustrated magnetic interactions onto the polarization system, potentially creating a new exotic electronic state, a polarization liquid.

Quantum spin liquids (QSL)[1–5], particularly those described by the exactly solvable Kitaev model[6] have attracted huge interest[7–12], due to their promising features for quantum computing, such as decoherence-protected quantum information computation, storage, and transmittance[13]. Unlike to geometrically frustrated models, the Kitaev model draws the frustration required for the QSL state from the bond-dependent interaction and not from the lattice geometry. The pristine Kitaev model was first sought after in Mott insulators with pseudospin $J_{eff}=1/2$, such as in materials with $Ir^{4+}$ and $Ru^{3+}$ ions on a honeycomb lattice[14–16], and recently materials with $Co^{2+}$ ions are also suggested as possible hosts[11]. However, as the long-range ordered magnetic ground states in all Kitaev candidates show the presence of significant additional exchange interactions[17,18], the realization of a pristine Kitaev model stays elusive, and so-far the long-range order could be suppressed only by the application of large magnetic field[15,16,19–22]. Still, these extended Heisenberg-Kitaev models have interesting consequences. The additional couplings were found to be important in the explanation of thermal transport phenomena[23,24], Raman scattering[25,26], and they lead to magnetic anisotropies and strong magnetoelastic effects[27,28]. It is an interesting question, if these

[1]Institut für Festkörperforschung, Leibniz IFW Dresden, Dresden, Germany. [2]Hochfeld-Magnetlabor Dresden (HLD-EMFL), Helmholtz-Zentrum Dresden-Rossendorf, Dresden, Germany. [3]International Center for Quantum Materials, School of Physics, Peking University, Beijing, China. [4]Beijing National Laboratory for Condensed Matter Physics, Institute of Physics, Chinese Academy of Sciences, Beijing, China. [5]Institute of Solid State and Materials Physics and Würzburg-Dresden Cluster of Excellence ctd.qmat, Technische Universität Dresden, Dresden, Germany. [6]Center for Transport and Devices, Technische Universität Dresden, Dresden, Germany. ✉e-mail: v.kocsis@ifw-dresden.de

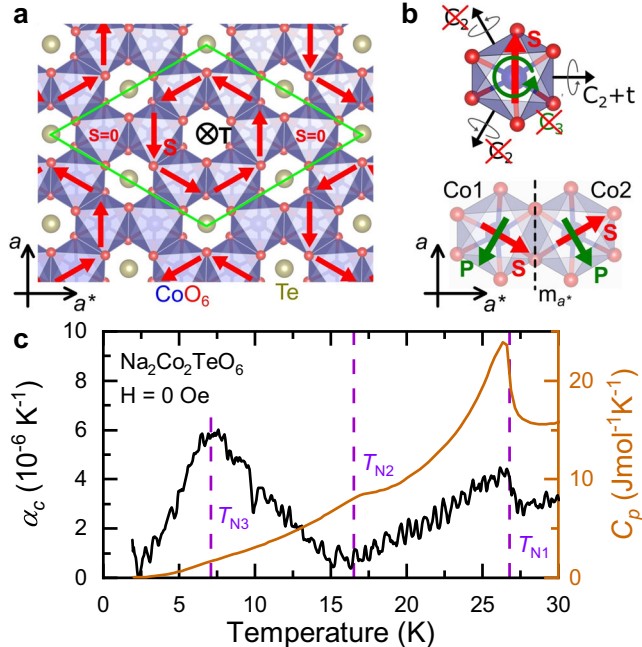

**Fig. 1 | Structural and thermodynamic properties of the quantum spin-liquid candidate Na$_2$Co$_2$TeO$_6$. a** Single honeycomb layer of CoO$_6$ octahedra with Te$^{6+}$ ions filling in the cavities. The red arrows (spins) illustrate the triple-q order with finite **T** ~ $\sum_i$**P**$_i$ × **S**$_i$ toroidal moment. **b** Local symmetries of a CoO$_6$ octahedra; the emergence of finite spin eliminates the C$_3$ and two of the C$_2$ symmetries, leaving only the C$_2$ symmetry perpendicular to $S$ combined with the $t$ time reversal symmetry. The broken symmetries allow the emergence of a local polarization ($P$) parallel to the Co-Co bond, which are anti-parallel to each other on the edge sharing CoO$_6$ octahedra, as these are connected by the $m_{a^*}$ mirror symmetry.
**c** Temperature dependence of the zero-field $\alpha_c$ thermal expansion coefficient and the $C_p$ heat capacity measured in $H$=0 kOe.

unwanted (but seemingly unavoidable) interactions can couple to external fields and can be further exploited in a completely different way to help the suppression of long-range order. Thus, recently, the application of uniaxial pressure[28] and electric ($E$) field[29–32] instead of magnetic field were suggested, as new control parameters for the stabilization of QSL phases, or for controlling the Majorana-Fermi surface[33], as theoretical possibilities. Moreover, the importance of magnetoelectric interactions was also pointed out as a possible explanation[29,30] for the strong electric dipole activity of THz excitations in d$^5$ transition metal Mott insulators[34] and in the Tunneling Spectroscopy of QSL materials[35]. While polar QSL materials have been reported, such as local polarization exists in $\alpha - RuCl_3$[36] and PbCuTe$_2$O$_6$ shows ferroelectricity[37], for long, magnetoelectric (ME) coupling was not considered possible in these $J_{eff}$=1/2 systems. In agreement with this expectation, so far no ME coupling has been found in any of the QSL candidates. However, only ME materials offer strong cross coupling between the electric and magnetic degrees of freedoms[38], and readily demonstrate $E$ control over magnetic ground states and magnetic excitations[39,40].

Here we provide the first direct evidence for a sizable ME and magnetoelastic coupling, and consequently, piezomagnetoelectric effects in a QSL candidate material. In order to find a QSL candidate material with ME coupling, we do a general survey among Kitaev magnets. We directly investigate both the ME and magnetoelastic properties of the QSL candidate Na$_2$Co$_2$TeO$_6$[41] with $J_{eff}$=1/2, in order to provide evidence for the origin of the ME effect. We find large magnetostriction and a sizable ME coupling, which greatly enriches the magnetic phase diagram and highlights the complex and anisotropic

magnetic interactions. We argue that the dominant source of the ME coupling in Na$_2$Co$_2$TeO$_6$ is the *pd*-hybridization mechanism[42,43], and therefore it can even host local $P$ independent from long-range magnetic order[44], while the spin-current mechanism can be also relevant. Our observation of simultaneous magnetoelasticity and magnetoelectricity demonstrates the presence of the piezomagnetoelectric effect. As an interesting consequence of the ME effect coexisting with the Kitaev interaction, we argue that an $E$ field could emerge as a new control parameter for the ground state and fractional spin excitations. The piezomagnetoelectric effect can offer an interesting experimental and theoretical ground for the investigation of coupled frustrated (magnetic) and frustration free (electric) systems endowed with a tuneable coupling parameter.

## Results

Due to the seemingly mutually excluding requirements of the ME effect and the Kitaev interaction, as discussed in the Supplementary Information, Na$_2$Co$_2$TeO$_6$ is unique among the QSL candidates. The chiral lattice (s.g. P6$_3$22) of Na$_2$Co$_2$TeO$_6$ is built up by co-aligned, edge-sharing CoO$_6$ octahedra pairs, which form a honeycomb layer with the Te$^{6+}$ ions filling in the cavities, as shown in Fig. 1(a). The honeycomb layers have alternating stacking, and are separated by Na$^+$ ions. The octahedral environment around the two non-equivalent Co$^{2+}$ ions has trigonal distortion[45], while the two neighboring CoO$_6$ are connected by a mirror symmetry ($m_{a^*}$, which is not a symmetry of the three-dimensional lattice). Each octahedra has a C$_3$ and three C$_2$ local symmetries (different from the lattice symmetries) with axes parallel and perpendicular to the $c$ axis, respectively. When the spin of the magnetic Co$^{2+}$ ion is aligned parallel to the $a$ axis (perpendicular to the Co-Co bonds), the local C$_3$ and two of the C$_2$ symmetries of the CoO$_6$ octahedron are broken. This allows the emergence of a local polarization ($P$) at both Co sites perpendicular to the spin and parallel to the axis of the preserved C$_2$ symmetry which is combined by the time reversal symmetry. In the paramagnetic phase, the absence of long-range order cancels the effect of the spin, effectively re-establishes the C$_3$ and C$_2$ symmetries, and precludes the emergence of macroscopic polarization, that is, Na$_2$Co$_2$TeO$_6$ is paraelectric. In the presence of an in-plane spin order, the effect of the spin on the local symmetries is not canceled, but even in this case the $m_{a^*}$ symmetry connecting the CoO$_6$ pairs renders these local $P$ antiparallel, i. e. the structure is antiferroelectric, as shown in Fig. 1(b).

At low temperatures, Na$_2$Co$_2$TeO$_6$ has an easy-plane antiferromagnetic (AFM) order with low anisotropy within the honeycomb plane[45–49]. The long-range AFM order develops at $T_{N1}$ = 26.8 K, which is followed by two further transitions at $T_{N2}$ = 16.5 K, and $T_{N3}$ = 7.1 K, which give additional complexity to the magnetic structure and indicate a canted planar AFM structure with weak-ferromagnetism[49]. At first, the magnetic ground state was described as a zigzag-type AFM order[45–48], however, recently a triple-q AFM state with a vortex-like spin order and possible phase co-existence is suggested, as shown in Fig. 1(a)[50–52]. Most recent theoretical models also suggest the emergence of composite multilinear order parameters, the so-called spin vestigial order above $T_{N1}$[53]. We emphasize, that only the triple-q AFM order allows the emergence of the ME effect due to the finite local polarization and the finite **T** ~ $\sum_i$**P**$_i$ × **S**$_i$ total toroidal moment of the unit cell[54].

Both our thermal expansion ($\alpha_c$-$T$) and heat capacity ($C_p$-$T$) measurements shown in Fig. 1(c) confirm these phase transitions. At $T_{N1}$, the thermal expansion and the heat capacity show clear anomalies, while at $T_{N2}$, $\alpha_c$ has a minimum and $C_p$ has a very slight anomaly. Finally, at $T_{N3}$, the thermal expansion measurement reveals a broad peak which is accompanied by only a weak anomaly in $C_p$. As the peaks in the thermal expansion are positive, according to the Ehrenfest relation[55] their uniaxial pressure dependence is also positive $\frac{\partial T_N}{\partial p_c} > 0$. Therefore,

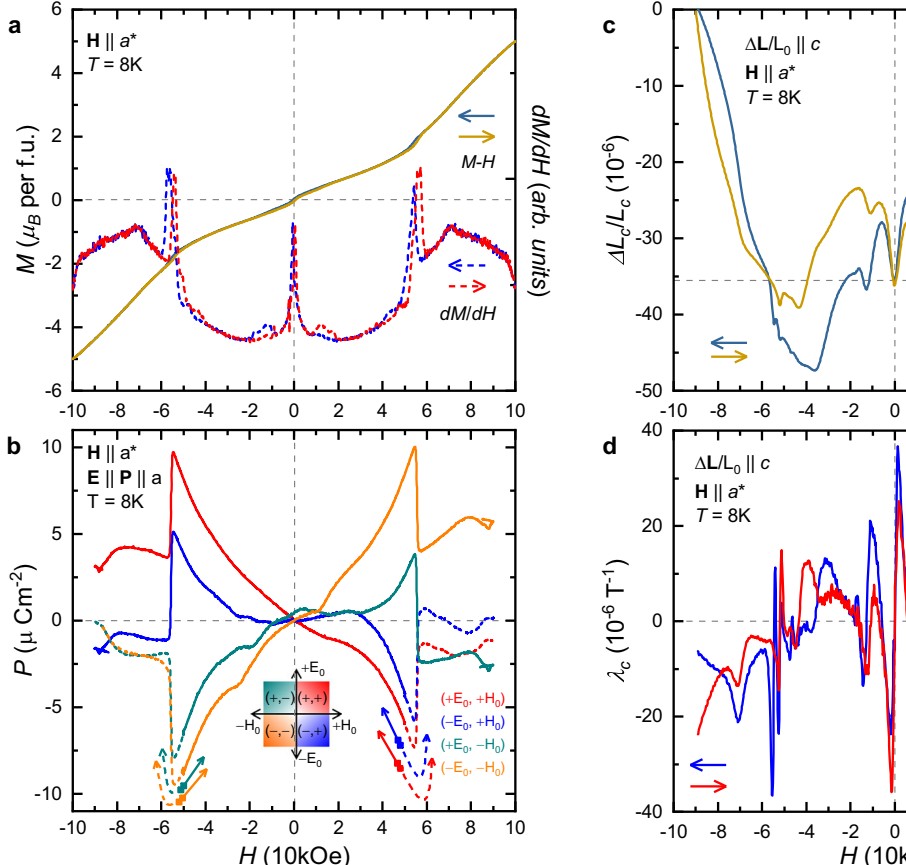

**Fig. 2 | Magnetic, magnetoelectric, and magnetoelastic properties of Na$_2$Co$_2$TeO$_6$. a** Magnetic field dependence of the $M$ magnetization (solid lines) and the $dM/dH$ field derivative (dashed lines) for **H**$\parallel a^*$ at $T$=8.0 K. **b** Poling-field dependence of the $H$-field induced in-plane polarization (**P**$\parallel a$) measured at $T$=8 K. The sample was cooled to the ordered phase in the presence of the four combinations of the poling fields ($\pm E_0, \pm H_0$), **E**$\parallel a$, **H**$\parallel a^*$. The measurements were started

from the $H_0$= ± 50 kOe poling field in the absence of $E$ field. Solid and dashed curves indicate measurements where the magnitude of the $H$ field is decreased or increased after poling, respectively. The $P$ shows mostly linear $H$-field dependence. **c** Magnetic field dependence of the measured $\Delta L_c/L_c$ relative length change ($\Delta L_c\parallel c$), and (**d**) the calculated $\lambda_c$ linear magnetostriction coefficient at $T$=8.0 K.

unlike to $\alpha$-RuCl$_3$[27], which has a negative peak in $\alpha_c^*$, in Na$_2$Co$_2$TeO$_6$ uniaxial pressure further stabilizes the AFM phase.

Figure 2 summarizes the field-dependent magnetic, magneto-electric, and magnetoelastic properties of Na$_2$Co$_2$TeO$_6$ measured at $T$=8.0 K for **H**$\parallel a^*$. First, we consider the magnetic and ME measurements shown in Fig. 2(a, b), respectively. In the low-field region, Na$_2$Co$_2$TeO$_6$ has a weak ferromagnetic moment, which is reversed at $H$=0 kOe and has an additional step-like increase at $H$=15 kOe. In the high-field region, we find a strong peak in $dM/dH$ at $H$=55 kOe in Fig. 2(a), which indicates the partial spin-flop like phase transition[48,49], while an additional weak peak emerges at $H$=75 kOe. Figure 2(b) shows the measured $P$-$H$ curves for all the four combinations of the poling fields ($\pm E_0, \pm H_0$), the ME poling procedure is described in the "Methods" section. The measured $P$-$H$ is the highest for the same sign combination of the ($E_0, H_0$) poling fields, and zero if $H_0$=0 kOe or $H_0$=90 kOe; as discussed in the Supplementary Information, a finite ME response is only observed if the sample is cooled in the presence of a moderate $H_0$ field. Na$_2$Co$_2$TeO$_6$ shows a nearly linear ME effect ($P = \chi^{ME}H$) with $\chi^{ME} = \mp 4$ ps/m maximum susceptibility, which is comparable to other $S$=1/2 systems, such as LiCu$_2$O$_2$[56] and SrCuTe$_2$O$_6$[57], but much smaller than in CoSe$_2$O$_5$[58]. At around $H$=55 kOe, the $P$ shows a sudden decrease and $\chi^{ME}$ drops to nearly zero, and the magnitude of the $\Delta P$=5 $\mu$C/m$^2$ change is comparable to that observed in PbCuTe$_2$O$_6$[37]. This drop in the $P$-$H$ curves coincides with the partial spin-flop like phase transition[48,49] as indicated by the peak-like anomaly in the $dM/dH$, as shown in Fig. 2(a).

The $\Delta L_c/L_c$ magnetostriction measured along the $c$ axis ($\Delta L_c\parallel c$) at $T$=8.0 K for **H**$\parallel a^*$, and the calculated $\lambda_c$ linear magnetostriction coefficient are shown in Fig. 2(c, d), respectively. Though in this configuration the magnetostriction measurements do not account for changes of lattice symmetry, $\lambda_c$ stays a sensitive indicator of magnetic and elastic phase transitions. The magnetostriction measurements reveal numerous anomalies for **H**$\parallel a^*$, suggesting a more complex phase diagram than that of $\alpha$-RuCl$_3$[27,59]. We note, that complementary magnetostriction measurements for both $\Delta L_a\parallel a$ and $\Delta L_{a^*} \parallel a^*$ have been reported in ref. 60. The anomalies in the $\lambda_c$-$H$ measurements coincide with features in the $dM/dH$-$H$ magnetization measurements. However, it is important to note, that the abundance of features in the $\lambda_c$-$H$ measurements is not reflected in the $P$-$H$ measurements, which implies that the origin of the ME effect is not related to magnetoelasticity. The $\lambda_c$-$H$ is neither symmetric nor anti-symmetric in $H$. Remarkably, the field increasing (↑, red curve) and decreasing (↓, blue curve) runs are connected by the $\lambda_c(H, ↑) = -\lambda_c(-H, ↓)$ symmetry. The $\lambda_c$-$H$ magnetostriction measurements show a wide hysteresis even up to the maximum $H$=90 kOe field with no corresponding hysteresis in the magnetization measurements.

Figure 3 shows the field induced polarization at selected temperatures for a complete cycle of the $H$ field. The ME poling procedure was the same as before with $E_0$=+1.5 kV/cm, $H_0$=+50 kOe fields and **E**$\parallel a$, **H**$\parallel a^*$. Measurements at higher temperatures show similar $P$-$H$ curves up to $T_{N1}$ as those shown in Fig. 2(b). When the $H$ field is

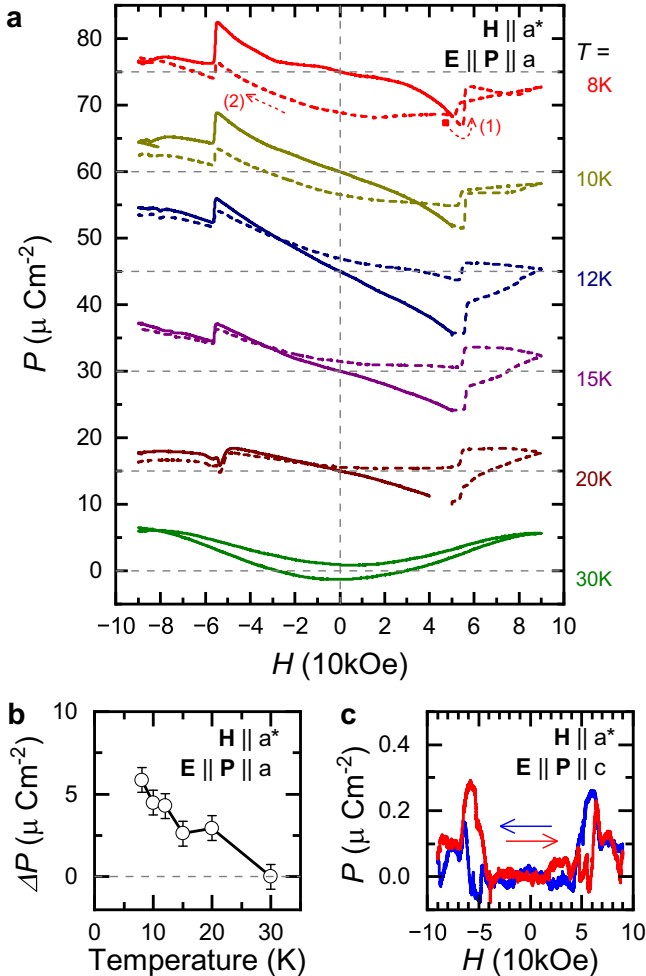

**Fig. 3 | Temperature dependence of the ME effect in Na$_2$Co$_2$TeO$_6$. a** Prior to the measurements the sample was cooled to the ordered phase in the presence of poling ($E_0$=+1.5 kV/cm, $H_0$=+50 kOe) fields, **E**∥$a$, **H**∥$a^*$, while measurements were started from $H$=+50 kOe in the absence of $E$ field. Dashed curves indicate measurements where the $H$ field was first swept to +90 kOe then reversed to -90 kOe, while in case of the solid curves the $H$ field was immediately swept towards -90 kOe. The $P$-$H$ loops were stacked along the vertical axis by 15 $\mu$C/m$^2$ for better visibility. The linear ME effect and the drop in $P$ at $H$=+55 kOe disappears above $T_{N1}$. **b** Temperature dependence of the $\Delta P$ polarization change at the spin-flop phase transition. **c** In-plane magnetic field dependence of the out-of-plane polarization measured at $T$=8 K (**H**∥$a^*$, **P**∥$c$).

swept through the partial spin-flop like phase transition[48,49] at $H \approx \pm 55$ kOe, $P$ shows a sudden drop ($\Delta P$) and $\chi^{ME}$ is strongly reduced. When the field is decreased again below $H$=55 kOe (see (1) and (2) in Fig. 3), the change in $P$ is significantly reduced, which suggests the emergence of a multi-domain ME state. In the paramagnetic phase, the linear ME effect and the drop in $P$ at $H$=+55 kOe disappear. Instead, $P$ is quadratic in field ($P \sim H^2$), which confirms the magnetic origin. Similarly to the $\lambda_c$ magnetostriction and to the $\epsilon_a$ dielectric constant measurements for **H**∥$a^*$ in ref. 60, the $P$-$H$ loops also show a hysteresis up to the highest $H$ field. Figure 3(b) shows the magnitude of the temperature dependence of the $\Delta P$ polarization drop measured at the spin-flop transition. The magnitude of $\Delta P$ gradually decreases with increasing temperatures, up to $T_{N1}$, where the spin-flop transition and $\Delta P$ disappear. The magnetic field dependence of the out-of-plane polarization (**P**∥$c$) at $T$=8 K is shown in Fig. 3(c). As expected due to lattice symmetry, the out-of-plane $P$ is almost zero at all fields and temperatures. However, in the field region of the spin-flop transition, the out-of-plane $P$ shows a small peak-like anomaly

with 20 times smaller magnitude than that of the in-plane $\Delta P$. This peak is not related to misalignment, as unlike the in-plane $P$, it is mainly symmetric to the reversal of the magnetization. Therefore, it is likely related to a non-collinear and non-planar spin structure transiently appearing in the spin-flop phase.

## Discussion

Using the combination of all heat capacity, magnetization, dilatometry, and polarization measurement data, we derive the magnetoelastic and ME phase diagram of Na$_2$Co$_2$TeO$_6$ for $H$∥$a^*$ field shown in Fig. 4(a). Further temperature and magnetic field dependent measurements are shown in the Supplementary Information. For **H**∥$a^*$, the phase diagram is significantly more complex than those of earlier reports[24,49,60]. Presumably, many of these new $H$-field-induced elastic transitions are related to changes in the interlayer stacking arrangement of the canted AFM or triple-q order along the $c$ axis (meaning the spin order within the $aa^*$ plane is shifted relative to that in adjacent layers) or to the rearrangement of the Na$^+$ ions due to shearing deformations, as illustrated in the supplementary information. To clarify these details, further neutron diffraction measurements are needed. We find evidence for a linear ME coupling in the field region $|H| \leqslant 55$ kOe (indicated by the purple area), while the decrease in $P$ for higher fields coincides with the appearance of the suggested and debated field-induced quantum spin liquid phase[41,61], which is turned into the field-induced ferromagnetic phase only at $H$=95 kOe. As we have directly observed both magnetoelasticity and magnetoelectricity, and the magnetic point group of the triple-q (P6$_3$2'2') order allows for the piezomagnetoelectric effect ($\mathbf{P}=\hat{\pi}\mathbf{H}\hat{\sigma}$, where $\hat{\pi}$ is the piezomagnetoelectric tensor and $\hat{\sigma}$ is the stress tensor)[62], we conclude that Na$_2$Co$_2$TeO$_6$ in the triple-q state is piezomagnetoelectric; i.e., uniaxial stress can enhance the ME effect, or magnetic (electric) field can allow the piezoelectric (piezomagnetic) effect.

We now turn to the origin of the ME effect. It is straightforward to exclude the exchange striction mechanism, the lattice is built up by uniformly charged Co$^{2+}$ ions, and thus there is no reason to expect dimerization within the $ab$ plane, needed for finite $P$, despite the sizable magnetostriction. Besides, $P$ of exchange striction origin should be quadratic in $H$ field at all temperatures and not linear as we observe here. If the exchange striction mechanism is significant, then the numerous sharp peaks in the $\lambda_c$-$H$ should also show up in $P$-$H$. In case of Na$_2$Co$_2$TeO$_6$, the spin-current mechanism can in principle explain the linear ME effect via either the off-diagonal exchange interaction (Dzyaloshinskii-Moriya) or via the ring-exchange interaction[53]. The Dzyaloshinskii-Moriya interaction leads to the canting of the spins, which can also explain the weak-ferromagnetic moment. However, upon the reversal of the weak ferromagnetic moment, $\chi^{ME}$ should also reverse, i.e., the $P$-$H$ curves should be again symmetric and above $T_{N1}$ the $P$ would be zero, and therefore we regard the spin-current mechanism less relevant. As discussed in Fig. 1(b), the highly distorted O$_6$ octahedra and the local symmetries of the crystal allows for the emergence of a local $P$ with an antiferroelectric order both in case of the orthorhombic zigzag and the trigonal triple-q phases. This is consistent with the $pd$-hybridization mechanism[42,43], where the hybridization between the magnetic ion $d$ and the ligand $p$ orbitals is affected by the orientation of the spin, leading to broken inversion symmetry, allowing for local polarization at the magnetic ion even independently from long-range magnetic order[44]. The distortion of the ligand cage around the Co$^{2+}$ ion allows finite $\mathbf{P}=\sum_\ell(\mathbf{S}\cdot\mathbf{e}_\ell)^2\mathbf{e}_\ell$ to appear in strong coupling to the spin, which gives a finite ME effect only in case of the triple-q order. As illustrated in Fig. 4(b, c), the triple-q order can show linear ME effect on the macroscopic scale due to the simultaneous canting of the $S$ and the local $P$, while the sign of the ME effect depends on the selection of the AFM domain. Besides, $pd$-hybridization is known to permit quadratic $H$-field dependence above $T_{N1}$, similarly to that of

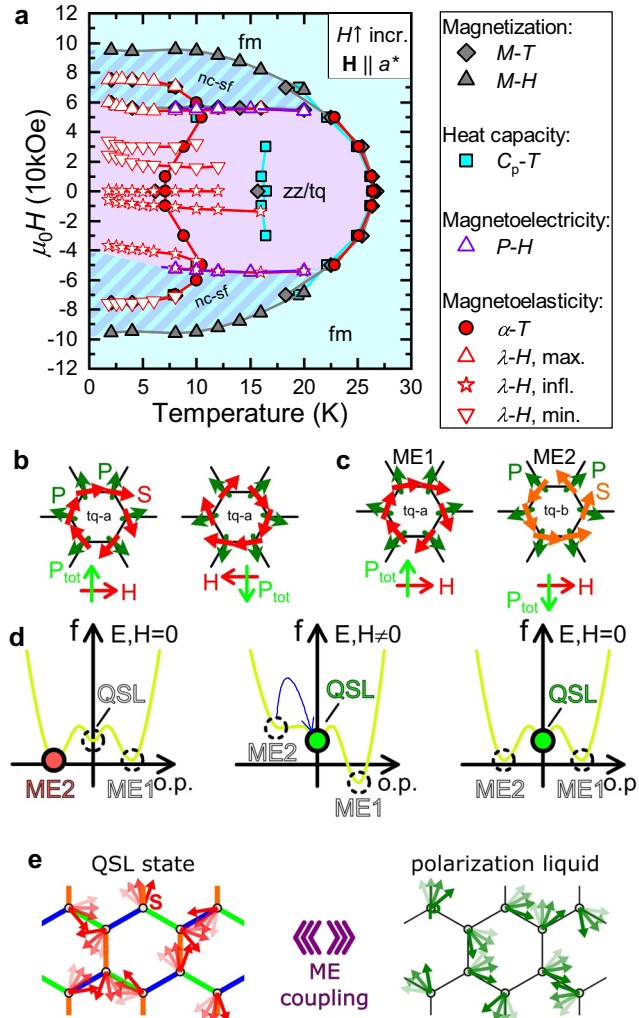

**Fig. 4 | The magnetoelectric and magnetoelastic *H-T* phase diagram of Na₂Co₂TeO₆. a** The phase diagram for **H**∥*a** field is based on all magnetization, heat capacity, dilatometry and ME measurements. Gray symbols correspond to magnetic, cyan symbols to heat capacity, purple symbols to *P-H*, and red symbols to $\lambda_c$ measurements. Colored regions indicate the zigzag/triple-q (zz/tq), the non-colinear spin-flopped (nc-sf), and the field-induced ferromagnetic (fm) phases. **b** In the triple-q (tq-a) phase the **P**ₜₒₜ total polarization of the unit cell is reversed for the reversal of the *H* field (linear ME effect). The $\chi^{ME}$ ME susceptibility is proportional to the **T** toroidic moment. **c** Sign of the ME effect is reversed when the other AFM state (tq-b) is selected. **d** Illustration of a free energy landscape with two stable ME states and a metastable QSL state, which is barely affected by the fields. The initial ME2 state is destabilized by the application of *E* and *H* fields. While the ME1 state is more stable, the system is trapped in the metastable QSL phase even when the fields are removed. **e** The *pd*-hybridization can raise a local polarization (**P**) around the magnetic ion even in the absence of long-range magnetic order. The ME coupling transfers the effect of the frustrated bond-dependent interactions of the spin system on the frustration-free polarization system to create a polarization liquid state.

Ca₂CoSi₂O₇[44]. Therefore we identify the *pd*-hybridization mechanism and the emergence of local *P* to be the dominant source of the ME effect in Na₂Co₂TeO₆, while the spin-current mechanism could in principle be relevant as well.

In summary, our direct measurements prove the simultaneous presence of magnetoelectric (ME) and magnetoelastic effects in Na₂Co₂TeO₆, which we conclude as the first example for a Kitaev QSL candidate material with piezomagnetoelectricity. Although our observations provide direct evidence for piezomagnetoelectricity, the nature and origin of the local polarization should be further tested by site-selective experimental methods, such as Second Harmonic

Generation, Nuclear Quadrupole Resonance, Resonant Inelastic X-ray Scattering. Using the thermodynamic, magnetic, ME, and magnetoelastic measurement data, we map out the phase diagram for **H**∥*a** featuring the newly discovered ME and magnetoelastic phase transitions. Based on the symmetries of the lattice and the *H* field dependence of the ferroelectric polarization, we point out the *pd*-hybridization mechanism as the dominant origin for the ME effect.

The coexistence of the ME effect with the Kitaev interaction can have interesting and important consequences motivating further experimental and theoretical research, as illustrated by two examples in Fig. 4(d, e). Inspired by earlier research on ME materials[39,63,64], the combination of *E* and *H* field could be used as new control parameters to stabilize a QSL ground state, as shown in Fig. 4(d). In a Kitaev magnet with coexisting stable ME and metastable QSL phases the application of *E* and *H* field can destabilize a preset ME state, while leaving the QSL state unchanged. During the switching from the instable to the stable ME states, the sample can get trapped in the metastable QSL phase. A similar interaction between ME and non-ME phases has been observed in Y-type hexaferrites[64]. On the other hand, in ME materials with the *pd*-hybridization mechanism, a local polarization (*P*) at the magnetic ion can emerge even in the absence of long-range magnetic order, which in principle may lead to a polarization liquid state to accompany the QSL state, as shown in Fig. 4(e).

## Methods

The platelet-shaped single crystals of Na₂Co₂TeO₆ were grown with a flux method described in refs. 48,49. The field-dependent magnetization and heat capacity data up to *H*=140 kOe were measured with a vibrating sample magnetometer and with the heat capacity option of the Physical Property Measurement System (PPMS, Quantum Design). Pulsed-field magnetization up to *H*=500 kOe was measured at the Hochfeld-Magnetlabor Dresden (HLD), using a compensated pickup-coil magnetometer in a ⁴He flow cryostat[65].

Thermal expansion and magnetostriction were measured using the PPMS-compatible Kuechler-mini dilatometer probe, which employs the capacitance measurement technique (AH2700A, Andeen-Hagerling)[66,67]. The length change was measured along the *c* axis (Δ*L_c*∥*c*), on a 190 μm-thick sample with 2.8 mm² area. During the thermal expansion measurement the temperature of the sample was swept at a constant 0.25 K/min rate, while in case of the magnetostriction measurement the magnetic field was swept with 10 Oe/s rate at constant temperatures. In case of the capacitance measurement technique a quasi-constant ~ 4 N force is applied on the sample by a spring, which equals to 1.4 MPa uniaxial pressure. We highlight that the application of uniaxial pressure cannot introduce changes in the lattice symmetry. The linear thermal expansion coefficient along the *c* axis ($\alpha_c$) was calculated as the temperature derivative of the relative length change:

$$\alpha_c = \frac{\partial}{\partial(T)} \frac{\Delta L_c(T,H)}{L_c(300\,\text{K},\,0\,\text{kOe})}, \tag{1}$$

where the $L_c(300\,\text{K},\,0\,\text{kOe})$ is the total thickness of the sample at room temperature. The linear magnetostriction coefficient ($\lambda_c$) was calculated as the *H*-field derivative of the relative length change:

$$\lambda_c = \frac{\partial}{\partial(\mu_0 H)} \frac{\Delta L_c(T,H)}{L_c(300\,\text{K},\,0\,\text{kOe})}. \tag{2}$$

The ferroelectric polarization was measured in the PPMS equipped with an electrometer (6517A, Keithley). The silver-paint electrodes were placed along the 190 μm-thick sides of the sample parallel to the *a** axis, i.e., **P**∥*a*, which provided a *S*=0.4085 mm² cross section for the measurement the distance of the electrodes was 1.6 mm. For the **P**∥*c* measurement, we used the same sample with silver-paint electrodes of *S*=0.9 mm² cross section. In case of the ME measurements, we applied

the following ME poling procedure. The sample was warmed to $T=50$ K prior to every $P$-$H$ cycle measurement, then cooled to the ordered phase in the presence of $E_0 = \pm 1.5$ kV/cm and $H_0 = \pm 50$ kOe fields ($\mathbf{E}_0 \| a$, $\mathbf{P} \| a$, $\mathbf{H}_0 \| a^*$), which is the so-called ME poling procedure. Following the poling, the $E$ field was removed and the leads were shorted for 10-minute-long intervals for a dozen times for electrostatic discharge, then $P$ was measured in sweeping $H$ field starting from $H_0$. This long discharging time was highly important, as due to the in-plane $Na^+$ ion conduction the sample acted as a discharging battery[68]. The ME measurements were done on the same crystal piece, which we earlier used for dilatometry measurements. The applied 1.4 MPa uniaxial pressure can therefore introduce substantial residual strain along the $c$ axis. As the $\pi_{aa^*cc}$ component of the piezomagnetoelectric tensor is finite, the $\sigma_{cc}$ uniaxial stress can introduce or enhance the $\chi_{aa^*} = \pi_{aa^*cc}\sigma_{cc}$ ME tensor element. Here we note that earlier measurements did not find ME effects in $Na_2Co_2TeO_6$[60]. The application of uniaxial stress prior to our ME experiments can explain why earlier investigations and measurements on samples without uniaxial pressure did not show ME effect.

The calculated $\hat{\chi}^{ME}$ and $\hat{\pi}$ tensors were cross-checked using the MTENSOR application[69], while magnetic ground states of various QSL candidates were taken from MAGNDATA[70] of the Bilbao Crystallographic Server.

## Data availability
The data presented in the current study are available from the corresponding authors on request and shared on public data repository ZENODO (https://doi.org/10.5281/zenodo.19397244).

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

## Acknowledgements

The authors are grateful to fruitful discussions with Izabella Lovas, Luis Elcoro, Satoshi Nishimoto, Xiaochen Hong, Christian Hemker-Heß, Lukas Janssen, and Matthias Vojta. Illustration of the structural unit cell was created using the software VESTA[71].

## Author contributions

The thermal expansion, magnetostriction, heat capacity and magnetization measurements were done by V.K. with the help of N.P. and A.U.B.W. The magnetoelectric measurements were performed by V.K., W.Y., and Y.L grew the single crystals. S.L. and H.K performed the pulsed-field magnetization measurements. A.U.B.W., V.K., and B.B conceived the project. V.K. wrote the manuscript with the help of Y.L., A.U.B.W., and B.B.; all authors contributed to the discussion of the results.

## Funding

We acknowledge financial support from the German Research Foundation (DFG) through the Individual Research Grant (project-id 540912241), the Collaborative Research Center SFB 1143 (project-id 247310070), and the Würzburg-Dresden Cluster of Excellence on Complexity, Topology and Dynamics in Quantum Matter ctd.qmat (EXC 2147, project-id 390858490). The work at Chinese Academy of Sciences and Peking University was supported by the National Basic Research Program of China (Grant No. 2025YFA1411501) and by the National Natural Science Foundation of China (Grant No. 12474138). We acknowledge the support of HLD at HZDR, member of the European Magnetic Field Laboratory (EMFL). V. K. was supported by the Alexander von Humboldt Foundation. Open Access funding enabled and organized by Projekt DEAL.

## Competing interests

The authors declare no competing interests.
