## [Transparent Peer Review file · Nature Communications]

Piezomagnetolectric effects in a candidate Kitaev magnet

Corresponding Author: Dr Vilmos Kocsis

Version 1:

Reviewer comments:

Reviewer #1

(Remarks to the Author)

Kitaev quantum spin liquid (QSL) is a promising candidate for topological quantum computation and has garnered significant attention in condensed matter physics. Most Kitaev materials exhibit a magnetically ordered state, and the application of external stimuli, such as pressure and magnetic fields, is a notable approach to suppress magnetic orders, potentially realizing Kitaev QSL. In this study, V. Kocsis and colleagues report the observation of the piezomagnetolectric effect in the Kitaev material $\text{Na}_2\text{Co}_2\text{TeO}_6$. These findings suggest the possibility of electric control over magnetic ground states and could offer a novel pathway to realizing Kitaev QSL in real materials.

The results presented are intriguing and contribute valuable insights into Kitaev materials. The article is well-written, demonstrating both high originality and the importance of the study. Therefore, I recommend its publication in Nature Communications.

However, before acceptance, a few points need to be addressed:

1. In the $E \parallel a$ configuration, a significant electric polarization up to $10 \mu\text{C}/\text{m}^2$ is observed in the P-H experiments. Could the authors provide temperature-dependent P (pyroelectric measurements) data? Additionally, how does the polarization behave in the $E \parallel c$ configuration?
2. The presence of electric polarization is linked to inversion symmetry breaking. It is advisable for the authors to cross-check the inversion symmetry breaking with other techniques, such as second harmonic generation, to strengthen their claims.
3. The authors suggest that exchange striction is unlikely to be the origin of the electric polarization due to the lack of correspondence between magnetostriction and electric polarization. However, the multiple transitions observed in the magnetostriction data are not entirely clear. The magnetization data shows considerably fewer transitions. Given the thin sample ($190 \mu\text{m}$) used in the c-axis magnetostriction measurements, there could be significant uncertainties, particularly if the capacitor plates are not perfectly parallel. It would be beneficial to stack a few samples along the c-axis to obtain more reliable c-axis magnetostriction measurements. Furthermore, could the authors provide data on the magnetoroton along the a-axis?
4. Several theoretical and experimental studies have also reported magnetoelectric effects and electric polarization in another Kitaev candidate, RuCl_3 [Phys. Rev. B 98, 125135 (2018); Phys. Rev. B 97, 161108 (2018); Phys. Rev. Lett. 125, 267206 (2020); Phys. Rev. Lett. 125, 227202 (2020); Phys. Rev. B 103, 134444 (2021); Sci. China-Phys. Mech. Astron. 67, 297511 (2024)]. The authors should consider citing these references appropriately to provide a broader context for their findings.

Reviewer #2

(Remarks to the Author)

Recently, the Kitaev antiferromagnets like $\alpha\text{-RuCl}_3$ and $\text{Na}_2\text{Co}_2\text{TeO}_6$ have received extensive research attentions for their interesting quantum magnetisms, including the QSL. In this work, Kocsis et al. reported the piezomagnetolectric effect in $\text{Na}_2\text{Co}_2\text{TeO}_6$, that is, the simultaneous presence of magnetoelectric and magnetoelastic effects. The authors concluded as the first example for a Kitaev QSL candidate.

In general, the experiments were well done and the results displays many new phenomena. This work is suitable for Nature Communications. However, I have some comments to be considered by the authors.

(i) The paper was not well written. Some tedious sentences are difficult to understand.

(ii) The right side of page 4. "The λ c-H magnetostriction measurements show a wide hysteresis even up to the maximum $H=90$ kOe field with no corresponding hysteresis in the magnetization measurements." Can authors give explanation about this result?

(iii) The right side of page 4. "Many of the H-field induced elastic transitions are presumably due to changes in the interlayer stacking of the canted AFM or triple-q order." It is difficult to understand the word and physics, which are quite important in this work.

(iv) The left side of page 6. "Inspired by earlier research on ME materials, the combination of E and H field could be used as new control parameters to stabilize a QSL ground state and to manipulate fractional spin excitations." Can authors give some explanation about the underlying physics in this sentence? It is probably a key issue of this work. How is the QSL stabilized by the combination of E and H field? How are the fractional spin excitations manipulated? Do the present experiments probe QSL state and the fractional spin excitations?

Version 2:

Reviewer comments:

Reviewer #1

(Remarks to the Author)

The authors have addressed my concerns, and I now believe the manuscript is suitable for publication in Nature Communications. However, I noticed a minor typo that should be corrected: in the last paragraph of page 4, the text reads, "The magnetic field dependence of the out-of plane polarization ($P \parallel c$) at $T=8K$ is shown in Fig.3(b)." This should refer to Fig. 3(c).

Reviewer #2

(Remarks to the Author)

The comments were well addressed and the revised manuscript can be accepted for publication in Nature Communications.

Response to the Reviewer comments

Reviewer #1 (Remarks to the Author):

Kitaev quantum spin liquid (QSL) is a promising candidate for topological quantum computation and has garnered significant attention in condensed matter physics. Most Kitaev materials exhibit a magnetically ordered state, and the application of external stimuli, such as pressure and magnetic fields, is a notable approach to suppress magnetic orders, potentially realizing Kitaev QSL. In this study, V. Kocis and colleagues report the observation of the piezomagnetolectric effect in the Kitaev material $\text{Na}_2\text{Co}_2\text{TeO}_6$. These findings suggest the possibility of electric control over magnetic ground states and could offer a novel pathway to realizing Kitaev QSL in real materials.

The results presented are intriguing and contribute valuable insights into Kitaev materials. The article is well-written, demonstrating both high originality and the importance of the study. Therefore, I recommend its publication in Nature Communications.

We sincerely thank Reviewer #1 for the careful reading of our manuscript, for the helpful comments, and we are pleased to learn that our results were found to be of sufficient interest for publication in Nature Communications.

However, before acceptance, a few points need to be addressed:

1. In the $E \parallel a$ configuration, a significant electric polarization up to $10 \mu\text{C}/\text{m}^2$ is observed in the P-H experiments. Could the authors provide temperature-dependent P (pyroelectric measurements) data? Additionally, how does the polarization behave in the $E \parallel c$ configuration?

To provide additional insight into the temperature dependence of the field-induced polarization, we have plotted the temperature dependence of the ΔP polarization jump in the new Fig. 3b. Unfortunately, direct temperature-dependent polarization measurements were unreliable due to high background currents, possibly related to the highly mobile Na^+ ions.

Motivated by the suggestion of Reviewer #1, we have performed polarization measurements in the $E \parallel c$ and $H \parallel a^*$ setting, which are shown in Fig. 3c. As expected, due to the lattice symmetry, the out-of-plane polarization is significantly weaker than the in-plane polarization. Interestingly, a peak appears in the P-H curve ($P \parallel c$ and $H \parallel a^*$) at the spin-flop transition. The peak is H-field symmetric; it shows hysteresis similar to that of the magnetization, but its magnitude is about 20 times smaller than that of the in-plane ΔP polarization jump. This peak is likely related to a non-collinear and

non-planar spin structure transiently appearing in the spin-flop phase; however, to reveal further details, more detailed neutron diffraction studies would be needed.

We modified the main text at the following points:

- We added figure panels Fig. 3c and 3c.
- We modified the main text:
‘Figure 3(b) shows the magnitude of the temperature dependence of the ΔP polarization drop measured at the spin-flop transition. The magnitude of ΔP gradually decreases with increasing temperatures, up to T_{N1} , where the spin-flop transition and ΔP disappear. The magnetic field dependence of the out-of-plane polarization ($P \parallel c$) at $T=8$ K is shown in Fig. 3(b). As expected due to lattice symmetry, the out-of-plane P is almost zero at all fields and temperatures. However, in the field region of the spin-flop transition, the out-of-plane P shows a small peak-like anomaly with 20 times smaller magnitude than that of the in-plane ΔP . This peak is not related to misalignment, as unlike the in-plane P , it is mainly symmetric to the reversal of the magnetization. Therefore, it is likely related to a non-collinear and non-planar spin structure transiently appearing in the spin-flop phase.’
- We added to the Methods section:
‘For the $P \parallel c$ measurement, we used the same sample with silver-paint electrodes of $S = 0.9\text{mm}^2$ cross section.’

2.The presence of electric polarization is linked to inversion symmetry breaking. It is advisable for the authors to cross-check the inversion symmetry breaking with other techniques, such as second harmonic generation, to strengthen their claims.

We agree with Reviewer #1 that, while our direct measurement method can provide irrefutable evidence for the magnetoelectric effect, the presence of local polarization and the microscopic origin requires local methods, such as second-harmonic generation, nuclear quadrupole resonance, or resonant inelastic X-ray scattering. However, such experiments would require a dedicated study and are beyond the scope of the present manuscript. To invoke the reader's attention to the limitation of our findings and to highlight this important direction for future work, we have added the following remark in the summary:

‘Although our observations provide direct evidence for piezomagnetolectricity, the nature and origin of the local polarization should be further tested by site-selective experimental methods, such as Second Harmonic Generation, Nuclear Quadrupole Resonance, or Resonant Inelastic X-ray Scattering.’

3. The authors suggest that exchange striction is unlikely to be the origin of the electric polarization due to the lack of correspondence between magnetostriction and electric polarization. However, the multiple transitions observed in the magnetostriction data are not entirely clear. The magnetization data shows considerably fewer transitions. Given the thin sample ($190 \mu\text{m}$) used in the c-axis magnetostriction measurements, there could be significant uncertainties, particularly if the capacitor plates are not perfectly parallel. It would be beneficial to stack a few samples along the c-axis to obtain more reliable c-axis magnetostriction measurements. Furthermore, could the authors provide data on the magnetoroton along the a-axis?

For magnetostriction and thermal expansion measurements, we use capacitance dilatometry, in which the sample is mechanically connected to a separate capacitor plate (the sample is not inside the capacitor). The non-parallelity of the capacitor plates can lead to nonlinear distortions in the measured magnetostriction near the highest measurable capacitances. Therefore, the maximum measurable capacitance, C_{MAX} , is a good indicator of the non-parallelism of the capacitor plates, and measurements must be conducted at capacitance levels well below C_{MAX} . Deviations resulting from non-parallel capacitor plates are accounted for in our calculations, based on the model of Pott's paper (R. Pott and R. Schefzyk, Journal of Physics E: Scientific Instruments 16, 44 (1983) citation [66]). In our dilatometer, the maximum capacitance is $C_{\text{MAX}} = 107 \text{ pF}$ (this is regularly tested before every measurement); we operate in the 20-25 pF range, and the capacitance change during the measurement is typically less than 0.01 pF. Using the Pott-Schefzyk method, we can calculate the effect of larger or smaller non-parallelities in the dilatometer's capacitor plates, as shown in the figure on the right. We approximate the ideal case with $C_{\text{MAX}} = 10^4 \text{ pF}$ (100 times larger than the actual case). We find that our measurements exhibit a maximum of 5% distortion relative to the ideal case, whereas significant distortion would only occur if the maximum capacitance fell below 50 pF. In conclusion, we are confident that the features observed in the magnetostriction measurements cannot be related to the parallelity of the capacitors.

Our dilatometry setup is ideal for measuring on thinner, plate-shaped samples. In this case, the sample geometry allowed us to measure length changes along the c-axis, but our setup cannot

measure length changes along the a - and a^* -axes. However, magnetostriction along both the a - and a^* -axes has already been measured by Zhang et al. (PRB, 2023), cited as Ref.[60], who used a different dilatometry method based on interferometry. To invoke the attention of the readers, we have added the following to the main text:

We note, that complementary magnetostriction measurements for both $\Delta L_a \parallel a$ and $\Delta L_{a^*} \parallel a^*$ have been reported in Ref. 60.

To provide more accurate statistics on the thickness dependence of our magnetostriction measurements, we have performed additional measurements on samples with different thicknesses, including a significantly thicker (280 μm) and an extremely thin (30 μm) sample (besides the original 190 μm thick piece). The magnetostriction measurements show very good reproducibility; all features are reproduced. However, when the 280 μm and 190 μm thick samples were stacked, some features, most notably the peak at 7.5 Tesla were not completely reproduced. We attribute this difference to a compensation effect of shearing deformations, which we explain in greater detail in the Supplementary Information in connection with Reviewer #2's comment.

We added Figure S4 and the following text to the supplementary:

In Fig. S4, we compare additional λc magnetostriction measurements performed on three different samples at $T=8\text{K}$ for $\Delta L_c \parallel c$ and $H \parallel a^*$. In Fig. S4(a), we repeat the λc - H data shown in Fig. 2(d) measured on the 190 μm thick sample. In Fig. S4(b) and S4(c), we show the λc - H data measured on a 280 μm and a 30 μm thick sample pieces, respectively. The reproducibility of the magnetostriction measurements is remarkably high, and detailed features in the λc - H data are well reproduced across all three samples, including the overall magnitudes of the peaks in λc . Most remarkably, although the signal-to-noise ratio is worst for the extremely thin 30 μm sample, we find that all phase transitions are reproduced within the measurement accuracy. However, when the two thickest samples are stacked on top of each other, as shown in Fig. S4(d), only the most prominent features at $H=0$ kOe and $H=\pm 55$ kOe are reproduced and finer details are lost. This difference may be related to slight sliding of the samples on each other, or a compensation effect of shearing deformations. We note that a similar stacking of crystals for dilatometry measurements was applied in Ref. [1].

4. Several theoretical and experimental studies have also reported magnetoelectric effects and electric polarization in another Kitaev candidate, RuCl_3 [Phys. Rev. B 98, 125135 (2018); Phys. Rev. B 97, 161108 (2018); Phys. Rev. Lett. 125, 267206 (2020); Phys. Rev. Lett. 125, 227202 (2020); Phys. Rev. B 103, 134444 (2021); Sci. China-Phys. Mech. Astron. 67, 297511 (2024)]. The authors should consider citing these references appropriately to provide a broader context for their findings.

We thank Reviewer #1 for bringing these relevant and important works to our attention. New publications were added as [29, 30, 31, 33, 35, 36], and we added Phys. Rev. Lett. 119, 227202 (2017) as [34]. We have complemented our introduction based on these publications:

Thus, recently, the application of uniaxial pressure [28] and electric (E) field [29–32] instead of magnetic field was suggested as new control parameters for the stabilization of QSL phases, **or for controlling the Majorana-Fermi surface [33]**, as theoretical possibilities. **Moreover, the importance of magnetoelectric interactions was also pointed out as a possible explanation [29, 30] for the strong electric dipole activity of THz excitations in d^5 transition metal Mott insulators [34] and in the Tunneling Spectroscopy of QSL materials [35]. While polar QSL materials have been reported, such as local polarization exists in α -RuCl₃[36] and PbCuTe₂O₆[37] shows ferroelectricity, for long, magnetoelectric (ME) coupling was not considered possible in these $J_{\text{eff}}=1/2$ systems. In agreement with this expectation, so far, no ME coupling has been found in any of the QSL candidates.**

Reviewer #2 (Remarks to the Author):

Recently, the Kitaev antiferromagnets like α -RuCl₃ and Na₂Co₂TeO₆ have received extensive research attentions for their interesting quantum magnetisms, including the QSL. In this work, Kocsis et al. reported the piezomagnetolectric effect in Na₂Co₂TeO₆, that is, the simultaneous presence of magnetolectric and magnetoelastic effects. The authors concluded as the first example for a Kitaev QSL candidate.

In general, the experiments were well done and the results displays many new phenomena. This work is suitable for Nature

We gratefully thank Reviewer #2 for carefully reading our manuscript and for the helpful comments. We are pleased that our manuscript has been deemed suitable for publication in Nature Communications.

Communications. However, I have some comments to be considered by the authors.

(i) The paper was not well written. Some tedious sentences are difficult to understand.

We thank Reviewer #2 for pointing out this issue and have revised the manuscript accordingly.

(ii) The right side of page 4. "The λ c-H magnetostriction measurements show a wide hysteresis even up to the maximum H=90 kOe field with no corresponding hysteresis in the magnetization measurements." Can authors give explanation about this result?

To provide a possible explanation, we have added a short description and Fig. S7 to the Supplementary. As shown in Fig. S7(a), while magnetization curves show negligible hysteresis, the magnetostriction, polarization, and dielectric constant measurements (Zhang et al. (PRB, 2023), cited as Ref.[60]) show sizeable hysteresis up to 140kOe. However, the hysteresis in the magnetostriction measurement differs from that of the magnetization, polarization, and dielectric constant measurements; The hysteresis in λ_c - H disappears at H = 0 kOe (while the hysteresis in magnetization is finite), which suggests that this apparent hysteresis at finite fields in the magnetostriction is a result of the linear piezomagnetic effect. Using group theoretical calculations, we find that such a linear piezomagnetic effect in Na₂Co₂TeO₆ is only allowed in the P632'2' magnetic space group, with such components that cause shearing deformation. While this type of deformation would not be measurable with our dilatometer, it is possible that the shearing deformation slides the adjacent Co₂TeO₆ layers relative to each other, thereby rearranging the Na⁺

ions within the aa^* plane and along the c axis (illustrated in Fig. 7(c)). Such a scenario may also explain the hysteresis in the P-H and dielectric constant curves; however, further measurements would be needed to confirm this explanation.

We added the following description and Fig. S7 to the Supplementary:

In Fig. S7, we discuss the hysteresis appearing in the magnetization and magnetostriction measurements. In Fig. S7(a), we show the magnetization measured at $T=8\text{K}$ for the field up (\uparrow , yellow curve) and down runs (\downarrow , blue curve), while the hysteresis is gauged by the difference, defined as $\delta M = (M_{\downarrow} - M_{\uparrow})/2$ (green curve, scaled for better visibility). The hysteresis is largest at $H=0$ kOe and $H=\pm 55$ kOe, due to the weak ferromagnetism and the spin-flop transition, respectively, and completely disappears above $H=\pm 60$ kOe. In Fig. S7(b), the magnetostriction measurements show a different picture. The hysteresis in the magnetostriction, which is defined similarly to that of the magnetization, $\delta L/L_0 = (\Delta L/L_{0,\downarrow} - \Delta L/L_{0,\uparrow})/2$, shows zero hysteresis at $H=0$ kOe. This means that the deformation is elastic and the crystal fully returns to its original elastic state when the field is removed, and the hysteretic part of the deformation is linear in magnetic field. A possible interpretation is illustrated in Fig. S7(c). In the hexagonal and piezo-magnetolectric $P632'2'$ magnetic space group, shearing deformations are allowed with the linear piezomagnetic effect: $\epsilon_{a^*c} = \lambda_{a^*ca^*} H_{a^*}$, where ϵ_{a^*c} is the a^*c component of the ϵ deformation tensor, $\lambda_{a^*ca^*}$ is one of the allowed piezomagnetic tensor elements, and H_{a^*} is the a^* component of the magnetic field. The ϵ_{a^*c} means such a deformation, where the top and bottom aa^* planes of the unit cell are slid along the a^* axis, which deformation component is not expected to be detected by our dilatometry measurements with $\Delta L_c \parallel c$. This is because in general $\Delta L = \mathbf{n} \epsilon \mathbf{n}$, where \mathbf{n} is the axis of the dilatometry measurement. However, the case of $\text{Na}_2\text{Co}_2\text{TeO}_6$ is particular, due to the distribution of the Na^+ ions within the unit cell. As a possible scenario, the shearing ϵ_{a^*c} deformation induced by the H_{a^*} magnetic field slides the Na^+ ions within the hexagonal plane, and redistribute them more evenly or slide them on top of each other, as illustrated in Fig. S7(c) with figures (1) to (4). Therefore, the redistribution of the Na^+ ions along the c axis caused by the shearing component of the linear piezomagnetic effect may provide an explanation for the observed hysteresis in our magnetostriction measurements. We note, that the displacement of the Na^+ ions may also explain the hysteresis in the P-H and dielectric constant curves, however, further measurements would be needed to confirm this explanation.

- (iii) The right side of page 4. "Many of the H-field induced elastic transitions are presumably due to changes in the interlayer stacking of the canted AFM or triple-q order." It is difficult to understand the word and physics, which are quite important in this work.

Following the advice of the Reviewer, and motivated by the explanation related by the previous remark, we have modified the main text:

Presumably, many of these new H-field-induced elastic transitions are related to changes in the interlayer stacking arrangement of the canted AFM or triple-q order along the c axis (meaning the spin arrangement within the aa^* plane is shifted relative to that in adjacent layers) or to the rearrangement of the Na^+ ions due to shearing deformations, as illustrated in the supplementary information. To clarify these details, further neutron diffraction measurements are needed.

- (iv) The left side of page 6. "Inspired by earlier research on ME materials, the combination of E and H field could be used as new control parameters to stabilize a QSL ground state and to manipulate fractional spin excitations." Can authors give some explanation about the underlying physics in this sentence? It is probably a key issue of this work. How is the QSL stabilized by the combination of E and H field? How are the fractional spin excitations manipulated? Do the present experiments probe QSL state and the fractional spin excitations?

In multiferroics, a single domain ME state (the order parameter) can be selected via the application of sufficiently high E and H fields. Once a multiferroic sample is stabilized in a ME state, it is possible to control the magnetic (electric) states using only electric (magnetic) fields.

In high-temperature multiferroics, such as in hexaferrites, one of the most severe issues hindering real-life applications is the loss of ME-state control (Kocsis, Nat. Commun. 2019; Kocsis, PRB 2020). For example, hexaferrites exhibit a complex magnetic phase diagram with multiple metastable magnetic phases. There the loss of ME-state control occurs during switching between two magnetic states: the material partially or fully enters a metastable non-ME state, and once it loses its ME properties, it cannot recover. While this is an issue to be addressed in multiferroics, it could be exploited as a useful feature in magnetoelectric Kitaev magnets, as illustrated in Fig. 4(d). In this sense, if a QSL state can be stabilized or destabilized, then fractional spin excitations could be turned on or off. However, the present experiments alone cannot probe spin excitations; therefore, we removed the reference to fractional spin excitations.

We expanded the description in the main text and we added Refs.[62,63] (Kocsis, Nat. Commun. 2019; Kocsis, PRB 2020):

Inspired by earlier research on ME materials [39, 62, 63], the combination of E and H field could be used as new control parameters to stabilize a QSL ground state, as shown in Fig. 4(d). In a Kitaev magnet with coexisting stable ME and metastable QSL phases the application of E and H field can destabilize a preset ME state, while leaving the QSL state unchanged. During the switching from the unstable to the stable ME states, the sample can

get trapped in the metastable QSL phase. A similar interaction between ME and non-ME phases has been observed in Y-type hexaferrites [63]. On the other hand, in ME materials with the pd-hybridization mechanism, a local polarization (P) at the magnetic ion can emerge even in the absence of long-range magnetic order, which in principle may lead to a polarization liquid state to accompany the QSL state, as shown in Fig. 4(e).